# Adiabatic topological photonic interfaces

Anton Vakulenko[1,5], Svetlana Kiriushechkina [1,5], Daria Smirnova [2], Sriram Guddala [1], Filipp Komissarenko[1], Andrea Alù [1,3], Monica Allen[4], Jeffery Allen[4] & Alexander B. Khanikaev [1,3] ✉

Topological phases of matter have been attracting significant attention across diverse fields, from inherently quantum systems to classical photonic and acoustic metamaterials. In photonics, topological phases offer resilience and bring novel opportunities to control light with pseudo-spins. However, topological photonic systems can suffer from limitations, such as breakdown of topological properties due to their symmetry-protected origin and radiative leakage. Here we introduce adiabatic topological photonic interfaces, which help to overcome these issues. We predict and experimentally confirm that topological metasurfaces with slowly varying synthetic gauge fields significantly improve the guiding features of spin-Hall and valley-Hall topological structures commonly used in the design of topological photonic devices. Adiabatic variation in the domain wall profiles leads to the delocalization of topological boundary modes, making them less sensitive to details of the lattice, perceiving the structure as an effectively homogeneous Dirac metasurface. As a result, the modes showcase improved bandgap crossing, longer radiative lifetimes and propagation distances.

Topological photonic materials enable a new class of optical modes—boundary states—with extraordinary characteristics[1,2]. These include topological resilience, which makes them insensitive to defects and disorder by suppressing backscattering[3–5]. Additionally, in systems with preserved time-reversal symmetry, e.g., spin-Hall and valley-Hall type photonic systems, and higher-order topological structures, topological phases rely on lattice/sublattice symmetries that allow light to be endowed with additional degrees of freedom. In turn, this enables unique opportunities to control topological photonic modes by acting on such synthetic pseudo-spins. Indeed, pseudo-spins have been successfully used for directional excitation and steering of electromagnetic waves in both spin-Hall[5–9] and valley-Hall[10–14] systems, their heterogeneous junctions[15], and in spin-full higher-order topological insulators[16,17].

Despite various successes in the experimental realization of photonic topological phases, it is clear that symmetry-protected topological phases cannot offer the same degree of robustness as systems with broken time-reversal symmetry, e.g., photonic analogs of quantum-Hall and Chern insulators[4,18–26]. Disorder that breaks the geometrical symmetry protecting the topological phase inevitably results in mixing of pseudo-spins, which leads to back-reflection of topological edge states. While patterning techniques are precise enough to minimize (not eliminate) such undesirable symmetry-breaking defects even for the case of nanophotonic systems, but still gapping of boundary modes emerges due to the reduction of the symmetry at interfaces. Specifically, one of the most widely used photonic designs proposed by Wu and Hu[6,27] is bound to exhibit a small but measurable band gap[28] near the crossing between the edge bands, due to the reduction of the $C_6$ symmetry that is responsible for the pseudo-spin formation. Similarly, in valley-Hall-like designs (including kagome lattice designs[29] studied below), boundary states do not truly cross topological band gap and exhibit gaps with bulk modes. These gaps clearly show the fragility of the topology in the symmetry-protected systems[30].

Another limitation of topological photonic systems is their radiative nature. Specifically, spin-Hall structures based on the Wu and

[1]Electrical Engineering and Physics, The City College of New York (USA), New York, NY 10031, USA. [2]ARC Centre of Excellence for Transformative Meta-Optical Systems (TMOS), Research School of Physics, The Australian National University, Canberra, ACT 2601, Australia. [3]Physics Program, Graduate Center of the City University of New York, New York, NY 10016, USA. [4]Air Force Research Laboratory, Munitions Directorate, Eglin AFB, FL, USA. [5]These authors contributed equally: Anton Vakulenko, Svetlana Kiriushechkina. ✉e-mail: khanikaev@gmail.com

Hu design[6] have topological boundary modes above the light cone. This makes them radiative, which limits their lifetime and propagation distance. In turn, this may hinder the viability of these structure in practical applications, such as to realize topological resonators and lasers. Therefore, many experimental realizations rely on valley-Hall type designs[13,31] with boundary modes that reside below the light cone or on non-leaky guided modes in arrays of ring resonators[3,32–36].

Here we introduce and demonstrate a method to address some of these limitations, mitigating radiative losses and reducing the role of symmetry reduction in topological systems by topological boundary profile engineering. Remarkably, the topological interface profiles have been recently of interest in condensed matter physics, including for boundaries of crystalline topological insulators[37], magnetic and topological insulators[38], topological insulator and crystalline insulators[39], as well as trivial and topological insulator[40], to name just a few. This further evidences the importance of the topological interface profile engineering in both condensed matter and photonic concepts. Pursuant to this goal, we consider a class of topological boundaries where synthetic gauge potentials vary adiabatically in lieu of the commonly used "sharp" domain wall that has abrupt transitions between topologically distinct domains.

## Results

### Adiabatic photonic spin-Hall boundaries

The benefits offered by adiabatic topological interfaces can be demonstrated by revisiting a spin-Hall design of photonic metasurfaces[7,28,41,42] based on the proposal by Wu and Hu[6]. The structure represents a slab of high-index dielectric material patterned into a triangular lattice of hexamers of holes. The equilateral triangular holes[7] with corners facing the center of the unit cell (insets in Fig. 1a) yield a complete topological band gap. We chose a 220-nm-thick silicon on insulator (SOI) as the material platform for numerical modeling and experimental demonstrations below.

When the distance between triangles in the neighboring unit cells is the same, the resultant "frustrated" (or unperturbed) structure exhibits a photonic band structure with two degenerate Dirac cones at the Γ-point for pseudo-spin-up and -down states, respectively, and the pseudo-spins of the structure correspond to two oppositely rotating

dipolar and quadrupolar modes. This doubly degenerate Dirac band structure can be understood as originating from the hexagonal symmetry of the frustrated lattice, which can be reduced to a photonic graphene model[43] with two Dirac cones at the K and K' points folded onto the Γ-point. The shrinkage or expansion of the hexamers can be achieved by bringing triangles in the unit cell closer or farther apart. This removes the frustration and leads to effective spin-orbital coupling (a synthetic gauge field). This coupling induces interaction of dipolar and quadrupolar bands and opens a complete photonic band gap for each pseudo-spins. The band gaps for the shrunken and expanded cases are trivial and topological, respectively. In accordance with the bulk-boundary correspondence, this leads to the formation of topological boundary modes at the domain walls between these two types of crystals. Figure 1 shows an example of step-like domain wall with sharp transition from trivial (shrunken) to topological (expanded) crystals, and confirms emergence and strong localization of the topological boundary mode to the interface. Changing the shape of the domain wall from sharp to a smoother square-root profile (Fig. 1a, middle row) causes the edge mode to be less localized at the center of the smoothed domain wall, which is seen in both the near- and far-field (Fig. 1b, c, middle rows, respectively). An even smoother adiabatic transition, with shrinkage and expansion following linear profile, leads to an even more delocalized edge state. Indeed, a simple analytical model based on the effective Hamiltonian with the mass term $m(x)$ and Dirac velocity $v_D$ predicts the mode in the form $\boldsymbol{\psi}(x,y) = \boldsymbol{\psi}(y) \exp\{-\frac{1}{v_D}\int_0^x m(x)dx\}$ (see Supplementary Note 1), which, for the case of linear mass-term profile yields Gaussian field distribution of the edge mode rather than exponential one for the constant mass term. However, the frequency dispersion of the edge states is largely unaltered by this change (Fig. 1d) with the exception of the small band gap at Γ-point between forward (spin-up) and backward (spin-down) edge modes, that are caused by breaking of the $C_6$ symmetry at the domain wall. This band gap reduces from $\Delta f/f = 2.4 \times 10^{-4}$ for the sharp domain wall profile to $\Delta f/f \approx 0.6 \times 10^{-4}$ for the linear profile in Fig. 1, where $\Delta f$ is the spectral width of the small band gap and $f$ is central frequency. This indicates a lower sensitivity of the edge states to violation of rotational symmetry at the domain wall, which can be explained by adiabatic preservation of local rotational

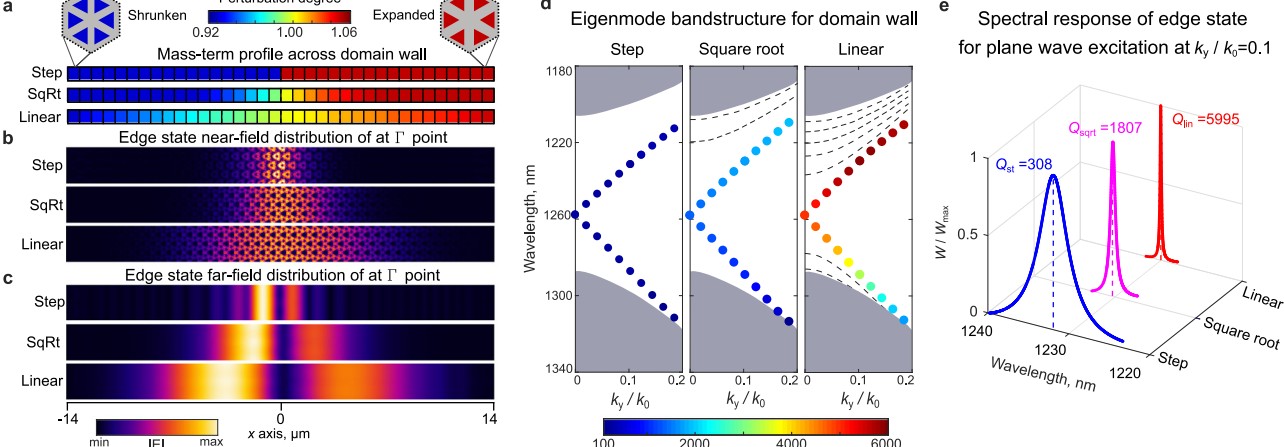

**Fig. 1 | Topological photonic boundary modes of adiabatic spin-Hall-type domain wall. a** Three profiles of the mass term (synthetic gauge field) with different degree of transition from trivial (shrunken) to topological (expanded) structures: non-adiabatic step-like, intermediate square-root, and adiabatic linear, respectively, from top to bottom. **b, c** Near-field and far-field profiles of boundary modes for the three profiles in (**a**) at the same frequency corresponding to the Γ-point (≈1260 nm). Far-field profile is taken one wavelength away from the surface of the structure. **d** Photonic band structure of topological boundary modes for the

step-like profile, and adiabatic square-root and linear mass-term profiles. Color shows quality factor *Q* of the edge modes. Gray shaded area corresponds to delocalized bulk modes and dashed lines show bulk states localized to the region of adiabatic transition (guided bulk modes[46]). **e** Simulations results for normalized energy density spectra $W/W_{max}$ of upper edge state excited by plane wave for the three types of domain walls. Note deliberately different degree of perturbation for shrunken and expanded profiles which is required to match the width of the photonic band gaps of topological and trivial domains.

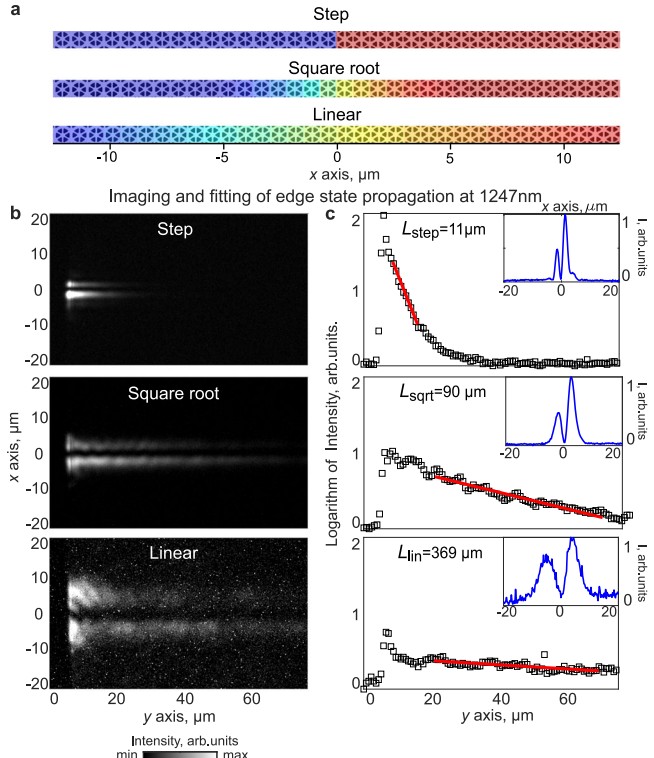

**Fig. 2 | Real-space images of edge states for sharp and adiabatic domain wall profiles. a** SEM images for the cross-section of domain walls with step, square-root and linear profiles. The perturbation degree of unit cells is color coded (shrunken/expanded domain is shown in blue/red). **b** Real-space images of edge states excited by spectrally filtered linearly polarized light. **c** Extracted intensity along the direction of propagation of edge states ($y$-axis) plotted on a natural log scale (square dots). Fitting of the data with the exponential decay yields the propagation length $L$ (red line). The inset shows the extracted profiles of the edge modes in the direction perpendicular to the propagation ($x$-axis).

symmetry across the smoothed domain wall. This advantage of an adiabatic topological interface can be qualitatively understood from basic group-theoretical considerations[30] applied to this specific structure, which predicts that the symmetry-generating pseudo-spin is well-defined only in the close vicinity of the Γ-point. If the mode is spatially less localized, e.g., in the direction perpendicular to the domain wall, then its distribution is more localized in the momentum along the same direction. Thus, the edge modes of the more adiabatic profile are expected to have better pseudo-spin characteristics.

Another limitation of the spin-Hall metasurface design stems from the leaky nature of bulk and edge modes because their spectrum lies above the light line. The magnitude of mode leakage increases as the degree of shrinkage or expansion grows[44]. This is a result of folding the Dirac cones from under the light cone into the proximity of the Γ-point. The larger deviation from the case of photonic graphene results in increased contribution to the radiative diffraction order[45]. The radiative decay of the modes introduces a tradeoff between the width of the topological band gap, which determines the operating bandwidth of the topological device, and the propagation distance of the edge states. We found that adiabatic domain walls allow to mitigate this problem, and that the radiative lifetimes can be increased by orders of magnitude due to the cancellation of the net dipole moment of the edge states (detailed analytical theory for this effect is provided in the Supplementary Note 1). In simple words, the dipole moments of evanescent tails of the edge modes in adjacent topological and trivial domains appear to be oppositely oriented (out of phase due to the 180-Zak phase). At the same time, more delocalized tails yield more

uniform near-field distribution giving rise to the far field with stronger directionality due to the reduction in the Fraunhofer diffraction (e.g., broader field profiles in real space have sharper momentum distribution). Therefore, slower attenuation of the edge mode for adiabatic profiles results in better cancellation of far fields from topological and trivial domains and an overall reduction in far-field radiation from the mode. For example, for the cases of step, square-root, and linear profiles considered in Fig. 1a, first-principles calculations for $k_y/k_0 = 0.1$ yield quality factors of $Q_{step} \approx 308$, $Q_{sqrt} \approx 1807$, and $Q_{linear} \approx 5995$, respectively. Figure 1e shows the reduced linewidth of the edge states obtained by a direct excitation of the edge states for the three cases and clearly reveals strong narrowing of the resonances.

Figure 1d shows the effect of the domain wall adiabaticity on the dispersion of the edge states and demonstrates that the spectral position of the modes remains nearly the same, and only the quality factors are affected by the transition. Note that, in addition to the delocalized bulk states in the topological and trivial domains, the adiabatic profiles lead to localization of some bulk modes to the transition region[46]. These modes increasingly penetrate the band gap region and effectively reduce the useable bandwidth of the topological modes. Nonetheless, the adiabatic profiles allow better control over the quality factor, and enable wider usable bandwidth for the edge states. For example, comparison of two profiles with the same usable bandwidth of 41 nm illustrates this point: (i) step profile with the degree of shrinking and expanding 2.4% and 2% respectively; and (ii) adiabatic linear profile with the maximum perturbations of 8% and 5%, respectively, and transition region of 39 cells, yields 2.5-fold increase in quality factor of the edge modes (see Supplementary Note 2).

We fabricated a set of samples realizing three domain wall profiles and measured their reflectivity spectra in a custom-built system capable imaging in both real and momentum (Fourier plane) spaces to experimentally verify our simulated prediction. The scanning electron microscopy (SEM) images for the cross-section of the three domain walls are shown in Fig. 2a. Figure 2b shows the far-field real space field profiles of the edge modes which matches well with Fig. 1c, and clearly shows both spreading of the modes perpendicular to the domain wall and increase in the propagation length $L$ of the modes. The increased propagation length is also confirmed by fitting the intensity of the reflection along the domain wall with an exponential function, which yields values of $L_{step} = 11 \, \mu m$, $L_{sqrt} = 90 \, \mu m$, and $L_{linear} = 369 \, \mu m$. This significantly extends the propagation distance of the edge modes while retaining their topological properties, as demonstrated in Fig. 3 where the reflectionless propagation around 120°-sharp bends is shown for the non-adiabatic step-like, intermediate square-root, and adiabatic linear domain wall, with the smoothest linear profile showing clear advantages. Figure 3a shows Fourier-plane images of the three cases under circularly polarized excitation and reveals both unidirectional edge states and bulk modes, with bands corresponding to edge modes with linewidth narrowing with increase in adiabaticity of the profiles. We note that a lower signal-to-noise ratio for smoother profiles that can be noticed in Figs. 2 and 3 is attributed to the reduction of radiative coupling of the edge states. As such, this reduction evidences the longer radiative lifetimes and higher radiative quality factors of the modes (see more details in Supplementary Note 4).

## Adiabatic valley-Hall boundaries in kagome lattice
Next, we consider a kagome photonic metasurface, which is known to exhibit a topological phase of Wannier type[47]. This design has been previously studied in the context of higher-order topological states that are supported in addition to edge modes[29,48]. The structure consists of three diamond-shaped holes in a high-index material arranged in a triangular lattice. Due to its hexagonal symmetry, the structure exhibits two Dirac cones at the two high symmetry points (K and K′ valleys) of the Brillouin zone. Similar to the spin-Hall structure discussed in the previous section, this structure can be shrunk or

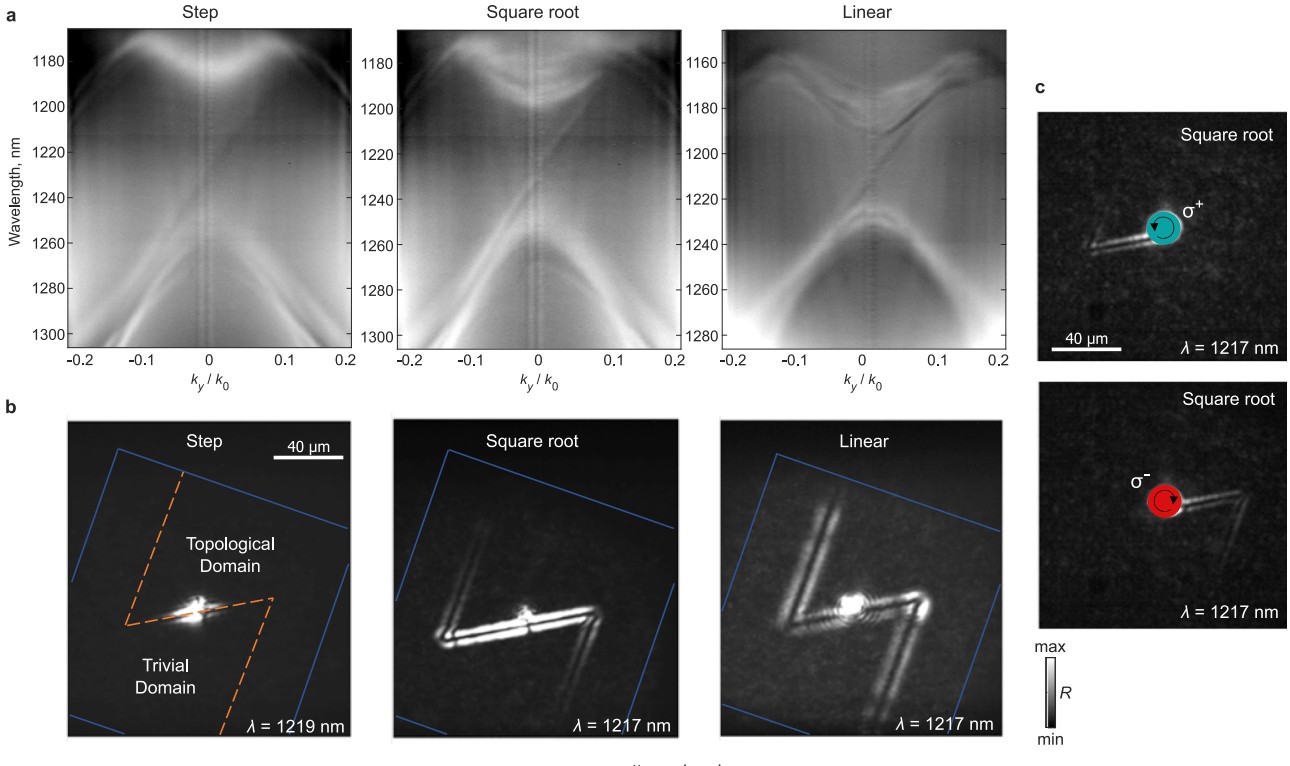

**Fig. 3 | Reflectionless and one-way propagation of edge states. a** Momentum-space (Fourier-plane) images showing dispersion of edge states for domain walls of different mass-term profiles. The states are excited by circularly polarized light. **b** Bidirectional propagation (linearly polarized excitation) of edge states along the domain walls with 120° bends. Images were taken in a cross-polarized configuration. **c** Example of unidirectional excitation of the edge states excited by the circularly left (right) polarized light $\sigma^+$ ($\sigma^-$) for the square-root mass-term profile.

expanded by moving the holes closer or farther apart from their neutral position. These perturbations lead to the formation of complete band gaps viz. topological for the expanded and trivial for the shrunken geometries, respectively[29,48]. Our design has holes in 220-nm-thick SOI substrate. In this case, the Dirac cones appear below the light line, rendering the associated modes non-radiative. This allows us to study the role of adiabatic variation of the mass term in the absence of radiative losses. In contrast to the spin-Hall structure, the kagome metasurface allows us to study the effect of adiabaticity on valley-polarized edge states in the absence of radiative decay.

Figure 4 shows the results of first-principle numerical simulations for the K-valley for both non-adiabatic step profile and adiabatic linear profile of the domain wall. The calculations were performed for two possible cuts along the Γ-K inclination (Fig. 4a, b upper insets). These two cases correspond to two opposite signs of the valley mass term and the valley Chern number that gives rise to the reversal of the slope of the valley edge states[10–12,49]. Similar to other valley designs, large values of the mass term (i.e., strong perturbation) lead to the gapping of the valley edge states with the bulk modes[50]. Similar to the spin-Hall design above, this gapping for our kagome design is associated with the reduction of lattice symmetries at the domain wall, therefore we expect that transition to the adiabatic domain wall will mitigate this effect. In fact, as seen from Fig. 4, the edge modes of linear domain wall completely close one of the gaps (upper for Cut I and lower for Cut II), while also reducing the second gap. The shape of the dispersion of the edge mode appears more linear for the adiabatic domain wall indicating a reduction in gapping of the edge states. We note that in addition to the edge modes, the adiabatic linear domain wall again hosts a new class of guided bulk modes trapped in the transition region from shrunk to expanded structures. These modes penetrate into the band gap but neither interact with the edge modes nor alter their dispersion.

The improvement of the edge mode properties, clearly seen from Fig. 4b, is associated with the broader real spatial profile of the edge modes and indicates a smaller span of modes in the momentum space. This makes them more localized near the range of momenta of a particular valley and renders the effective "homogenized" response of the structure closer to that of continuous Dirac equation. Thus, the valley edge states of the adiabatic system represent a better approximation to the Jackiw-Rebbi states induced by the reversal of the mass term[51].

We confirmed our theoretical predictions with a set of samples fabricated in the 220-nm-thick SOI substrates. The bulk and edge dispersion was measured using a custom-built Fourier-plane imaging system with integrated solid immersion phase matching scheme. This setup (details in Methods) can excite evanescent modes that reside below the light cone with one objective and collect their "leaked" field with another objective. A broad range of wavenumbers accessible with this setup allowed us to reconstruct the photonic band structure of the samples below the light cone. The experimental results, shown in Fig. 4, confirm the improved spectral characteristics of the edge states. The gaps between edge and bulk modes are either completely closed or reduced with overall straightening of the bands corresponding to the edge states.

## Discussion

In summary, we studied adiabatic topological interfaces produced by a smooth variation of the synthetic gauge fields in photonic metasurfaces. Two types of structures, spin-Hall Wu and Hu and valley-Hall kagome designs were theoretically investigated and the findings were experimentally verified. Our studies show that the profile of topological domain walls represent an excellent control parameter for various characteristics of topological edge states. Specifically, for the

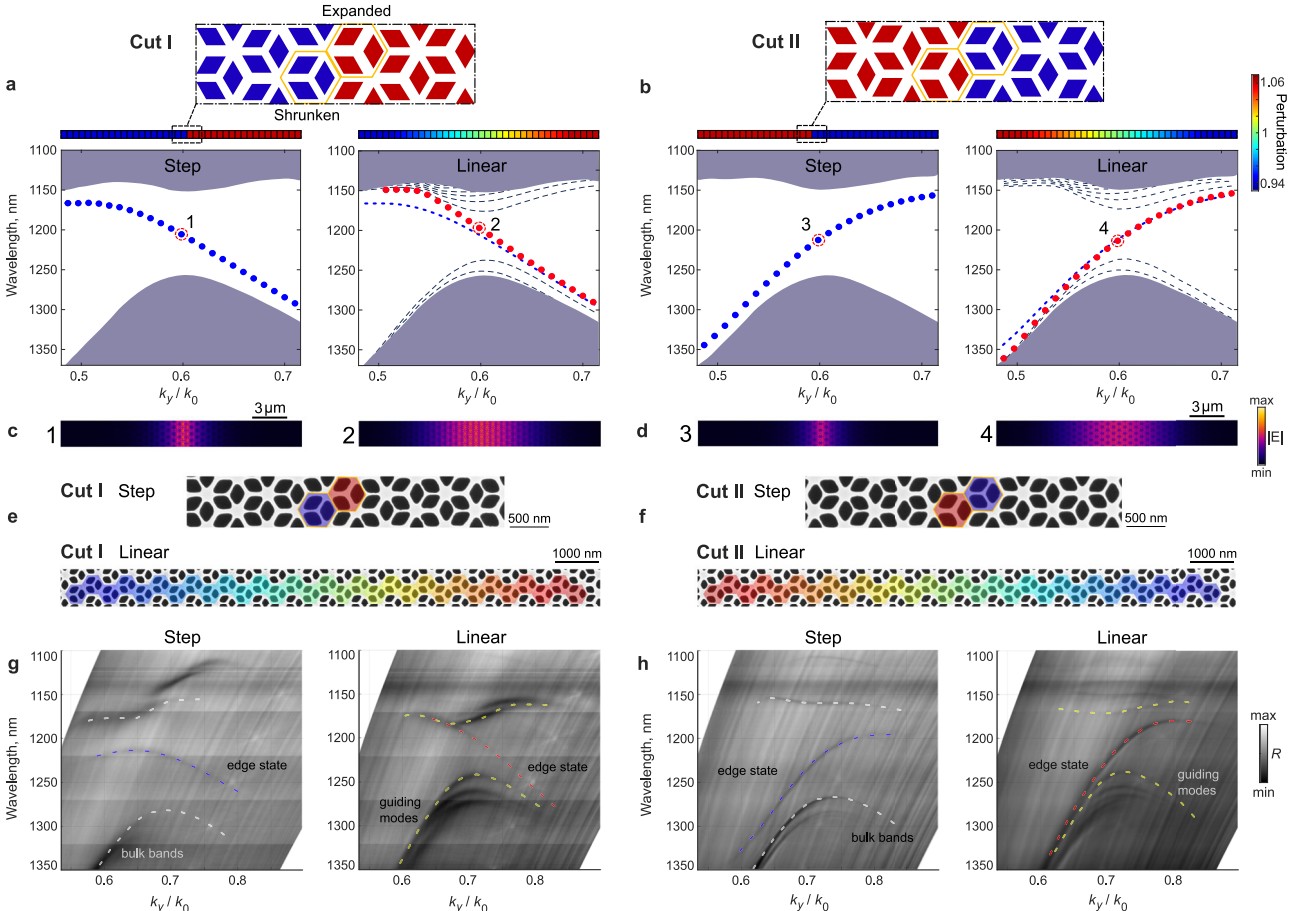

**Fig. 4 | Topological photonic boundary modes of adiabatic valley-Hall-type domain walls. a, b** Calculated band diagrams of topological photonic kagome metasurface with (i) step and (ii) adiabatic (linear) domain walls. Color-coded schematics of the corresponding domain walls of the two cuts are shown as insets. Dispersion of edge modes for step and linear domain wall profiles are shown by blue and red dots, respectively. Smaller blue dots corresponding to the edge state dispersion for the case of the step profile is provided for comparison in the case of adiabatic profile. Gray shaded area corresponds to delocalized bulk modes and gray dashed lines show bulk states localized to the region of adiabatic transition. **c, d** Near-field profiles of edge states for step and linear domain walls taken near the mid-gap for the two cuts. **e, f** SEM images of fabricated metasurfaces with step and linear domain wall for the two cuts. **g, h** Experimental momentum-space (Fourier-plane) images showing dispersion of edge states under the light cone for step and linear domain walls corresponding to both possible cuts. Additional colored dotted lines indicate boundary and bulk states.

spin-Hall design that supports leaky topological edge modes, we show that radiative properties of the modes can be controlled and improved e.g., significantly extending propagation distance of the modes. In both designs, the adiabatic variation of the mass term provides additional control over the bandwidth of the topological edge modes and reduces undesirable spectral gaps, including due to the local defects (Supplementary Note 5). These characteristics of the topological modes are relevant to many practical applications.

Our study offers a fresh perspective on the topological photonics from the point of view of "homogenized" continuous Dirac limit rather than traditional view based on "discrete" photonic crystal and topological chemistry pictures centered around symmetries of photonic orbitals. Our results pave the way to expanded opportunities for improved photonic devices, such as topological lasers and topological polaritons, where improved quality of modes is critical for device operation.

## Methods
### Sample fabrication and experimental details
Both spin-Hall and valley-Hall designs were fabricated on SOI wafers with 220-nm-thick Si film. Patterns were defined by electron beam lithography (Elionix ELS-G100) followed by etching (Oxford PlasmaPro) with inductively coupled plasma (ICP). Patterns were written on the spin-coated film of electron beam resist ZEP520A-7 (200 nm-thick) covered by a layer of anti-charging agent DisCharge H2Ox2. After development of the resist in n-amyl acetate during approximately 30–40 s the silicon layer under the exposed area was vertically etched by ICP (500 W) with the recipe based on the mixture of $C_4F_8$ (47 sccm), $SF_6$ (13 sccm) and $O_2$ (15 sccm) gases. The etch rate was 130 nm/min.

Imaging the spectral dispersion and real-space propagation of edge states in metasurfaces with spin-Hall-like design was carried out using a custom-built near-infrared microscope. Light from a halogen lamp was collimated and focused to the sample surface using a long working distance 50× microscope objective (BoliOptics 50×, 0.42 NA). The back focal plane of the objective was imaged in 4f configuration using the combination of a tube lens and a Fourier lens on to the entrance slit of the spectrometer (SpectraPro-HRS500, Teledyne Princeton Instruments). The dispersion from 300 g/mm grating was imaged by the NIR camera (NIT HiPe SenS 640) mounted to the exit slit of the spectrometer. A pair of linear polarizers and quarter wave plates were used in the optical path of incident light for the circular polarized excitation measurements. We used a laser beam with a linewidth of 5 nm for the real space imaging of the edge state propagation. The laser beam was generated by a supercontinuum light-source Leukos Electro-VIS with connected Leukos Tango-NIR2 acousto-optic tunable

filter. Image of real space propagation of the excited edge state was captured by the NIR camera.

Modes of the valley-Hall type metasurface based on kagome lattice appear under the light cone. We studied these modes using an Otto-like phase matching scheme using ZnSe hemispherical lens set up in close proximity of the sample to couple to the near-field of the modes. The experimental setup was adapted for solid immersion spectroscopy by adding a path for sample illumination at oblique incident with 50× objective, the original optical path was used for collection of light, and the sample holder was modified with a build-in mount for hemispherical lens. Similar approach was used in ref. 29, but the modified system allows Fourier-plane images of the sample to be collected which reveal a photonic band structure. The adjusted excitation path with the proximity of the K point of the band structure was probed by the excitation light which came from one objective at 55° angle of incidence to the sample. Reflected light coming out at −55° angle was collected by another objective, and then projected by lenses on spectrometer slit.

### Numerical simulations

We used commercial software COMSOL Multiphysics for the first-principles simulations presented in this work. The metasurfaces with sharp and adiabatic variation of mass term were simulated using an array of 39 unit cells along the direction of the mass-term profile variation (the $x$-axis). The periodic boundary conditions were applied in the $y$-direction. All other boundaries were surrounded by perfectly matched layers. In the eigenmode simulations (Fig. 1d), the quality factor $Q$ of each mode was calculated from the complex-valued eigenfrequency $\omega_c$ as $Q = \text{Re}(\omega_c)/2\text{Im}(\omega_c)$. In frequency domain simulations with the incident plane-wave source (Fig. 1e) the quality factor was evaluated from the resonant spectral response as $Q = \omega_0/\Delta\omega$, where $\omega_0$ is the central resonant frequency, and $\Delta\omega$ is a spectral width of the peak at half maximum.

### Data availability

The data that support the findings of this study are available from the corresponding author upon reasonable request.

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

## Acknowledgements
J.A. and M.A. would like to thank the AFRL/RW Chief Scientist Innovative Research Funds, and the Emerging Technologies funds. A.A. and A.B.K. acknowledge support from the Simons Foundation. A.B.K. acknowledges support from the Office of Naval Research (ONR) (N00014-21-1-2092) and from the National Science Foundation (NSF) (DMR-1809915). D.S. acknowledges support from the Australian Research Council (DE190100430, CE200100010). Fabrication of samples for this work was performed at the Nanofabrication Facility at the Advanced Science Research Center at The Graduate Center of the City University of New York.

## Author contributions
A.B.K. conceived the research. D.S. performed theoretical calculations. A.V. and S.K. performed first-principle simulations, fabrication of samples and optical characterization, including real space imaging and angle-resolved reflectivity measurements. A.V., S.G. and F.K. assembled the experimental setup. A.B.K., A.A., M.A. and J.A. guided and supervised the project. All authors participated in the discussions of the results and the manuscript preparation.

## Competing interests
The authors declare no competing interests.
