## [Peer Review File · Nature Communications]

REVIEWER COMMENTS

Reviewer #1 (Remarks to the Author):

I enjoyed reading this paper and the results achieved here, and would like to recommend the publication of this manuscript because it does provide a sound solution to the practical problems in topological photonics. As shown by the convincing experimental results here, the quality factor and the propagation length are significantly improved by introducing linearly modulated domain walls. However, there is still a question I would like to raise and wish to see the authors give some answer in this direction. The question is: whether the approach raised here is general (with some theoretical proof from possibly synthetic dimensions or gauge fields) or case by case?

Reviewer #2 (Remarks to the Author):

The authors theoretically and experimentally studied three different interfaces (step, parabolic, and linear types) in spin-Hall and valley-Hall topological photonic structures which are protected by geometrical symmetry. Because of fragility of symmetry-protected topology and the radiative and bosonic natures of photonic systems, topological boundary states at the abrupt (step) interface are less robust and may be partially broken. However, the authors found that adiabatic (linear) interfaces are superior to the abrupt and parabolic interfaces in the aspects of less sensitivity to details of the lattice, narrower linewidth, longer radiative lifetime and propagation distance. The adiabatic interface may be used to improve photonic devices, such as topological lasers and topological polaritons.

Considering the unique properties of photonics, to the best of my knowledge it is the first time that the authors compare different topological interfaces in topological photonics and clearly point out the advantages of adiabatic topological interface. In general, the authors have provided some interesting, fresh, novel and concrete results, and the manuscript is well-written and easy to follow. I believe that this manuscript could potentially benefit the practical applications of topological photonics and trigger more theoretical and experimental studies. However, at this stage, I could not convincingly recommend the acceptance of this manuscript in the current form, for the following reasons.

#1 Topological boundary states, as key features of topological state of matter, have attracted broad research interest. In the community of condensed matter physics, many studies have considered and compared different type of topological interfaces including abrupt and smooth types; see some references [J. Liu, et al., Phys. Rev. B 88, 241303(R) (2013); S. V. Eremeev, et al., Nano Lett. 18, 6521-

6529 (2018); T. V. Menshchikova, et al., Materials 13, 4481(2020); J. Bermejo-Ortiz, et al., arXiv:2207.09292 (accepted by Phys. Rev. B)] and the references therein.

However, the authors did not mention the advances in the introduction at all. The authors need to do systematical investigation, introduce some important developments in this field, and discuss the differences and relations between their results and the existing findings.

#2 The authors did some case studies to show how the adiabatic topological interface is superior to the step and parabolic topological interfaces. However, as the width of linear topological interface changes from 0 to infinity, the topological interface also changes from abrupt to adiabatic types. To better understand the crossover mechanics, the authors should show how the width of linear interface affect the behaviors of topological boundary states.

#3 Related to the second comment, I do not believe the more adiabatic connection between two different topological structures, the linear topological interface has better behaviors in most circumstances. There may be an optimal width of linear topological interface that the topological boundary states have features of longer lifetime, propagation distances, and better topological protection again disorder. Consider an extreme case where the width of linear interface is infinite, the spatial width of topological boundary state is also infinite. I believe that any tiny disorder could break the topological boundary states. However, the step and parabolic interfaces are still immune to disorder to some extent. The authors should objectively evaluate the pros and cons of the adiabatic topological interface.

#4 The authors mainly discuss the topological interfaces in symmetry-protected topological systems. It is very interesting to consider the topological interfaces in photonic analogues of quantum-Hall and Chern insulators which are more robust. Could the authors give some intuitive discussion and comment on the advantages of adiabatic topological interface in these systems?

#5 The expression $\Delta f/f \approx (0.4 - 0.6) \times 10^{-4}$ in the line 94 is ambiguous.

Reviewer #3 (Remarks to the Author):

In their manuscript “Adiabatic topological photonic interfaces”, the authors introduce the concept of adiabatic topological photonic interfaces. Their aim it to demonstrate theoretically and confirm experimentally that adiabatic topological metasurfaces with slowly varying synthetic gauge fields significantly improve the guiding features of spin-Hall and valley-Hall topological photonic structures, which are commonly used in the design of symmetry- protected topological devices.

This is a very interesting manuscript; it will enrich the field as it has a very good story and very good experiments. I recommend this work for publication in Nature Communications.

For the sake of clarity, there are a few points the authors may answer/optimize before publication:

- Why is the SNR seemingly worse for these new types of interfaces (se Fig. 2b linear)
- How much light is left in the states?
- The authors may add a numerical analysis about how much more robust the states at the novel interfaces are against symmetry breaking (as this was part of the motivation)
- Figs. 4g,h are hard to decipher...
- Why are these interfaces called “adiabatic” and which role does it play?
- Why does this idea do not work for electronic TI and, more generally, for all SPT phases?

In conclusion, while I think the paper is already fine for publication, addressing the points above may serve for even more readability.

Authors' Response to Reviewers

Reviewer #1 (Remarks to the Author):

I enjoyed reading this paper and the results achieved here, and would like to recommend the publication of this manuscript because it does provide a sound solution to the practical problems in topological photonics. As shown by the convincing experimental results here, the quality factor and the propagation length are significantly improved by introducing linearly modulated domain walls. However, there is still a question I would like to raise and wish to see the authors give some answer in this direction. The question is: whether the approach raised here is general (with some theoretical proof from possibly synthetic dimensions or gauge fields) or case by case?

Authors' response to Reviewer #1:

We thank Reviewer #1 for emphasizing importance of our results in demonstrating practical solutions of existing problems in topological photonic structures. We truly appreciate their recommendation to publish our work in Nature Communications.

As for their question about whether the approach used by us is general, the answer is yes. The proof is as follows. The localization of the modes to the topological boundary is defined by the gauge field profile in the metasurfaces. Thus, for the mass term of a generic profile $m(x)$, where x is the direction perpendicular to the domain wall, the edge mode localization to the wall ($m(x=0)=0$) is given by the following expression, $\psi(x, y) = \psi(y) \exp\left\{-\frac{1}{v_D} \int_0^x m(x) dx\right\}$, where $m > 0$ ($m < 0$) for $x > 0$ ($x < 0$), v_D – Dirac velocity. As the result, for square root ($m(x) = \pm \frac{1}{v_D} bx^{1/2}$) and linear ($m(x) = ax$) profiles of the mass term, the localization is not the conventional exponential localization, but

$\psi(x, y) \sim \exp\left\{-\frac{2}{3v_D} bx^{3/2}\right\}$ and a gaussian like $\psi(x, y) \sim \exp\left\{-\frac{1}{2v_D} ax^2\right\}$, respectively. Thus, modulation of the gauge field allows one to control field distribution in the direction perpendicular to the domain wall. This theoretical model is confirmed by our numerical simulations and experiments. Both simulations (see Fig. R1 below) and experiment also confirm this prediction that the field becomes less localized for smoother profiles, with the linear case yielding the least localized boundary mode among the tested profiles. Importantly, responding to the Referees' question, this simple analytical form of the mode profile is generic and applies to any topological system, e.g., valley, spin-Hall, or QHE-like systems, and therefore the mass term modulation does in general allow to control properties of the boundary modes, such as their localization to the topological boundary.

In the context of the Wu and Hu structure specifically, the observed increase of the quality factor is the consequence of fact that the modes become less localized to the interface for smoother mass term profiles. Indeed, it is known that radiative lifetime of topological edge modes increases for less localized modes, which has been demonstrated for the step-like profiles with smaller mass term by Hafezi's group in Refs. [S. Barik, et al. A topological quantum optics interface. *Science* **359**, 666-668 (2018) (Supplementary Material); C. Flower, et al. Topological Edge Mode Tapering (2023) <https://doi.org/10.48550/arXiv.2206.07056>]. In our systems, however, this is achieved by different means – the mass-term profile modulation, but the result is the same, although with the benefit of larger topological band gap.

The mechanism for increased radiative lifetime (and quality factor) by itself is explained as a consequence of cancellation of the far fields radiated by two (trivial and topological) domains. Thus, because the respective fields are out-of-phase, this leads to better cancellation of the far-field radiation of the boundary mode when the radiating mode is less localized to the domain wall. The mode that is more spread out spatially into the bulk gives rise to more directional far field radiation. Therefore, for smoother mass term profiles, the far fields from the two domains are better aligned with one another, which, considering their out-of-phase character and equal amplitudes, gives rise to stronger cancellation and overall suppression of the radiative leakage into far field. Interestingly, this out-of-phase character of the

far-fields of topological and trivial domains always manifests itself in experiments as the dark (nodal) line in the images of the edge modes at the center of the domain wall (also seen in Refs. [S. Arora, et al. *Phys. Rev. Lett.* **128**, 203903 (2022), and S. Peng, et al. *Phys. Rev. Lett.* **122**, 117401 (2019)]).

Figure R1. Simulation results for far field distributions of electrical field E_x of the edge mode for domain walls with different profiles. Field distribution is plotted in the xy -plane $1.5 \mu\text{m}$ above the metasurface.

We have reflected the above discussion to the revised version of the manuscript (in the main text and in Supplementary Note 1) in response to the Reviewer #1 comment.

Reviewer #2 (general comment):

The authors theoretically and experimentally studied three different interfaces (step, parabolic, and linear types) in spin-Hall and valley-Hall topological photonic structures which are protected by geometrical symmetry. Because of fragility of symmetry-protected topology and the radiative and bosonic natures of photonic systems, topological boundary states at the abrupt (step) interface are less robust and may be partially broken. However, the authors found that adiabatic (linear) interfaces are superior to the abrupt and parabolic interfaces in the aspects of less sensitivity to details of the lattice, narrower linewidth, longer radiative lifetime and propagation distance. The adiabatic interface may be used to improve photonic devices, such as topological lasers and topological polaritons.

Considering the unique properties of photonics, to the best of my knowledge it is the first time that the authors compare different topological interfaces in topological photonics and clearly point out the advantages of adiabatic topological interface. In general, the authors have provided some interesting, fresh, novel and concrete results, and the manuscript is well-written and easy to follow. I believe that this manuscript could potentially benefit the practical applications of topological photonics and trigger more theoretical and experimental studies. However, at this stage, I could not convincingly recommend the acceptance of this manuscript in the current form, for the following reasons.

Authors' response to Reviewer #2 general comment

We would like to thank Reviewer #2 for carefully reading our manuscript and for highlighting novelty of our work and the advantages of the approach used for topological photonics. We also thank Reviewer #2 for pointing out ways to further improve our work by addressing their specific remarks raised, which all were addressed in the revised manuscript (detailed in our response below).

Reviewer #2 specific remark #1:

Topological boundary states, as key features of topological state of matter, have attracted broad research interest. In the community of condensed matter physics, many studies have considered and compared different type of topological interfaces including abrupt and smooth types; see some references [J. Liu, et al., *Phys. Rev. B* **88**, 241303(R) (2013); S. V. Eremeev, et al., *Nano Lett.* **18**, 6521-6529 (2018); T. V. Menshchikova, et al., *Materials* **13**, 4481(2020); J. Bermejo-Ortiz, et al., arXiv:2207.09292 (accepted by *Phys. Rev. B*)] and the references therein.

However, the authors did not mention the advances in the introduction at all. The authors need to do

systematical investigation, introduce some important developments in this field, and discuss the differences and relations between their results and the existing findings.

Authors' response to Reviewer #2 remark #1

We would like to thank Reviewer #2 for pointing out all these important developments in condensed matter physics, which escaped our attention, but are indeed directly relevant to our own work in photonics. Specifically, the choice of boundary, which is truly vital for crystalline topological phases, was investigated in Phys. Rev. B 88, 241303(R). Similarly, interfacing magnetic and topological insulators was studied in Nano Lett. 18, 6521-6529 (2018), and an advantage of smooth connections between magnetic insulators and topological insulators over sharp interfaces was discovered as a trivial boundary state appeared to be suppressed. While different from what we report in our work, the idea of interface engineering for improved performance of topological systems is strikingly similar to our own. Even more relevant is the work [Materials 13, 4481(2020)], where a smooth interface between topological insulator and topological crystalline insulator is created by epitaxial immersion growth, which leads to improved topological boundary modes. Finally, another very interesting discovery reported in [Phys. Rev. B 107, 075129] is that smooth heterojunctions of trivial and topological insulators lead to emergence of coexistence of massless Weyl and massive Dirac fermions, which further evidences importance of the topological profile engineering in both condensed matter and photonics.

Following Reviewer #2 suggestion, the respective advances brought by the topological interface engineering in condensed matter physics were mentioned in our revised manuscript and properly cited.

Reviewer #2 specific remark #2:

The authors did some case studies to show how the adiabatic topological interface is superior to the step and parabolic topological interfaces. However, as the width of linear topological interface changes from 0 to infinity, the topological interface also changes from abrupt to adiabatic types. To better understand the crossover mechanics, the authors should show how the width of linear interface affect the behaviors of topological boundary states.

Authors' response to Reviewer #2 remark #2

This is an interesting point which indeed deserves an additional elaboration in the manuscript. There are two main consequences of the transition from abrupt to adiabatic interface. Firstly, the boundary modes become progressively less localized. Thus, for the specific case of linear profile, with the mass term $m = ax$, the Dirac equation admits an exact edge mode solution, which shows that the modes are not exponentially localized to the interface but exhibit the Gaussian profile $\psi(x, y) \sim \exp\left\{-\frac{1}{2v_D} ax^2\right\}$. Thus, for the case of infinitely wide linear region, $a \rightarrow \infty$, the modes too become completely delocalized, i.e., effectively become one of the bulk modes. Secondly, some of the bulk modes also start to localize to the linear transition region, and, as its width approaches infinity, there is an infinite number of such modes which continuously populate the former gap region. Thus, the physics in the limit $a \rightarrow \infty$ becomes indistinguishable from the case of massless $m = 0$ and gapless Dirac equation. This behavior was confirmed in our numerical modelling by directly solving Maxwell's equations with the use of the finite element method, as shown in Fig. R2.

To summarize, when optimizing the smoothness of the mass-term profile, one should consider tradeoffs and aim for the scenario when the edge modes are still well-defined, i.e., exist within a reasonably wide topological bandgap, but are still sufficiently localized to the interface as required by a particular application.

The above discussion and results investigating dependence on the width of the linear profile were added to the revised Supplementary Information as a new Note 3.

Fig. R2. Photonic band structure of topological boundary modes for linear mass-term profile with gradually increasing width of transition (a-d) from 0 unit cells up to 37 unit cells. Color-coded diagrams of the degree of perturbation and edge mode near field distribution for each interface are shown on the top panels. Radiative quality factor of the edge modes and guided bulk modes are shown in color in these band diagrams.

Reviewer #2 specific remark #3:

Related to the second comment, I do not believe the more adiabatic connection between two different topological structures, the linear topological interface has better behaviors in most circumstances. There may be an optimal width of linear topological interface that the topological boundary states have features of longer lifetime, propagation distances, and better topological protection again disorder. Consider an extreme case where the width of linear interface is infinite, the spatial width of topological boundary state is also infinite. I believe that any tiny disorder could break the topological boundary states. However, the step and parabolic interfaces are still immune to disorder to some extent. The authors should objectively evaluate the pros and cons of the adiabatic topological interface.

Authors' response to Reviewer #2 remark #3

We thank Reviewer #3 for noting the existence of both pros and cons of adiabatic interfaces. Indeed, this is 100% true that progressively delocalized modes, while less sensitive to local defects and symmetry reductions due to the spreading of the field profiles (mode volume is simply much larger than the defect cross-section), also become less spectrally isolated due to the penetration of the bulk modes localized to the transition region into the band-gap region. As such, the presence of such trivial bulk modes at the same frequencies enables scattering between topological and these trivial states, thus limiting bandwidth of topologically resilient behavior. One therefore needs to carefully consider practical aspects of such limited bandwidth of adiabatic profiles and choose them according to their specific applications and needs. The need of finding "tradeoff" was explained in supplementary note 3 of the supplement. We also note our new results on robustness of edge modes corresponding to different mass term profiles to defects, which were encouraged by the Reviewer's comment and were added to the revised Supplement as a new Note 5.

Reviewer #2 specific remark #4:

The authors mainly discuss the topological interfaces in symmetry-protected topological systems. It is very interesting to consider the topological interfaces in photonic analogues of quantum-Hall and Chern

insulators which are more robust. Could the authors give some intuitive discussion and comment on the advantages of adiabatic topological interface in these systems?

Authors' response to Reviewer #2 remark #4

This is an interesting question. First, it is clear that the chiral boundary modes in the time-reversal symmetry broken photonic systems (mentioned by the Reviewer #2) too will exhibit spreading and will be less localized to an interface. This follows from the same effective Dirac description of these states (i.e., see description in S. Raghu and F. D. M. Haldane *Phys. Rev. A* **78**, 033834 (2008)). However, since in this case modes cannot exhibit backscattering because of the lack of any backward edge modes, we cannot expect them to be even more resilient to defects with respect to back scattering into such the modes. On the other hand, there will be trivial bulk modes localized to the transition region which penetrate increasingly into the band gap region for smoother profiles, which will enable scattering into these modes by defects and will ruin robustness. Thus, from the first glance, we do not see any advantages of adiabatic interfaces for systems with broken TR symmetry for photonic systems, but this question may be worth a separate study with a specific photonic system in mind.

Reviewer #2 specific remark #5:

The expression $\Delta f/f \approx (0.4 - 0.6) \times 10^{-4}$ in the line 94 is ambiguous.

Authors' response to Reviewer #2 remark #5

We thank Reviewer #2 for pointing out the ambiguity. The expression was revised.

Reviewer #3 (general comment):

In their manuscript "Adiabatic topological photonic interfaces", the authors introduce the concept of adiabatic topological photonic interfaces. Their aim it to demonstrate theoretically and confirm experimentally that adiabatic topological metasurfaces with slowly varying synthetic gauge fields significantly improve the guiding features of spin-Hall and valley-Hall topological photonic structures, which are commonly used in the design of symmetry-protected topological devices.

This is a very interesting manuscript; it will enrich the field as it has a very good story and very good experiments. I recommend this work for publication in Nature Communications.

Authors' response to Reviewer #3 general comment

We would like to thank Reviewer #3 for their very positive evaluation of our work and for recommending it for publications.

Reviewer #3 specific remark #1

Why is the SNR seemingly worse for these new types of interfaces (se Fig. 2b linear)

Authors' response to Reviewer #3 remark #1

The reason for the lower signal to noise ratio (SNR) is in the reduction of the far-field radiation originating from the edge modes corresponding to the smoother mass term profiles, which is also responsible for the longer radiative lifetimes of the modes and increase of radiative quality factors reported in the paper.

Thus, getting back to the original question of Reviewer #3, the lower SNR by itself is a confirmation of improved radiative properties of the boundary modes. We have clarified this point in the main text explaining reduction in the SNR for completeness:

"We note that a lower signal-to-noise ratio for smoother profiles that can be noticed in Figs. 2 and 3 is attributed to the reduction of radiative coupling of the edge states. As such, this

reduction evidences the longer radiative lifetimes and higher radiative quality factors of the modes (see more details in Supplementary Note 4).”

In the revised version, inspired by Reviewers' comments, we have also added a detailed explanation of the mechanism of the radiative loss suppression in the main text and Supplementary Note 1 (e.g., Fig. S2 and accompanying text). The explanation in the main text read as follows:

“In simple words, the dipole moments of evanescent tails of the edge modes in adjacent topological and trivial domains appear to be oppositely oriented (out of phase due to the 180-Zak phase). At the same time, more delocalized tails yield more uniform near-field distribution giving rise to the far field with stronger directionality due to the reduction in the Fraunhofer diffraction (e.g., broader field profiles in real space have sharper momentum distribution). Therefore, slower attenuation of the edge mode for adiabatic profiles results in better cancellation of far fields from topological and trivial domains and an overall reduction in far-field radiation from the mode.”

Reviewer #3 specific remark #2

How much light is left in the states?

Authors' response to Reviewer #3 remark #2

This is an interesting question. Indeed, as the radiative coupling of the edge states is getting smaller, it is becoming harder to couple incident light into these modes. Thus, one might expect less energy in the edge states for smoother profiles. However, the reduction in coupling efficiency is also partially compensated by the increased quality factors of the edge modes. The simplest description of these two competing mechanisms can be made in the framework of the coupled mode theory [Haus, H. A. Waves and fields in optoelectronics (1984)], which allows one to estimate the mode amplitude as

$$A = \frac{i\alpha}{\omega - \omega_0 + \frac{i}{\tau}}$$

where α is the radiative coupling efficiency of the mode, which is related to the radiative lifetime τ_R as $\alpha = \sqrt{1/\tau_R}$, $\tau = (\tau_R^{-1} + \tau_0^{-1})^{-1}$ is the total lifetime of the edge mode, and τ_0 is the lifetime of the mode due to all other processes, including absorption. Assuming resonant excitation, we see that the energy stored in the mode is given by

$$W \sim |A|^2 = \tau_R \tau_0^2 / (\tau_R + \tau_0)^2.$$

In the limit of very large τ_R this yields $W \sim \tau_0^2 / \tau_R$, i.e., the energy captured by the mode from the incident radiation will drop inverse proportionally to the radiative lifetime of the mode. At the same time, the reflectance drops even faster $R = W / \tau_R \sim \tau_0^2 / \tau_R^2$. Thus, although we might see very little power outflow from the edge mode (resulting in lower SNR), there is still a significant amount of energy stored in the mode.

We note, however, that the coupling from the far field is not the only way to excite boundary modes, and for very high radiative quality factors (as well as for most practical purposes), one would rather use coupling via grating integrated into the structure.

This discussion about the energy stored in the edge mode and the reflectance due to its excitation was added to the revised Supplement as a new Note 4.

Reviewer #3 specific remark #3

The authors may add a numerical analysis about how much more robust the states at the novel interfaces are against symmetry breaking (as this was part of the motivation)

Authors' response to Reviewer #3 remark #3

Following Reviewer #3 recommendation, we have performed numerical modelling and analyzed the effect of a local (symmetry-reducing) defect on the boundary modes. Introduction of such a defect is known to give rise to the pseudo-spin flipping, coupling between oppositely propagating edge states, and, eventually, gapping of the boundary modes. Thus, to understand the effect of the defect, we have calculated bandwidth of the gap between oppositely propagating modes for mass term profiles of different adiabaticity. We found that, indeed, due to the more spread field distributions for smoother profiles, edge modes were increasingly less sensitive to the defect, and the size of the defect-induced gap was respectively smaller. The width of the gap was calculated as the function of the symmetry reduction strength for three profiles of interest, showcasing the strongest robustness of the least localized linear mass term profile (Fig. R3).

Fig. R3. Simulations results for bandwidth of the defect induced gap at the Γ point for different mass term profiles (as indicated by the legend) for increasing local crystalline defect perturbation. Defect perturbation and its location at the interface (white rectangle) are shown alongside the mass term profiles in lower panel. The defect represents the symmetry reduction at one unit cell adjacent to the domain wall within the supercell. The average dielectric constant of the defective unit cell was kept the same (the volume fraction of air holes is maintained), which ensures that the center of the gap remain at the same spectral position.

We note, however, that if one considers the effect of disorder in the system, i.e., many defects distributed randomly over the structure, both sharp and adiabatic profiles will be similarly affected.

The respective discussion (almost exactly paraphrasing the response above) was added as a new Supplementary Note 5. The corresponding results were also mentioned in the main text.

Reviewer #3 specific remark #4

Figs. 4g,h are hard to decipher...

Authors' response to Reviewer #3 remark #4

We thank Reviewer #3 for pointing this out. We have revised Fig. 4 to make panels (g) and (h) more legible. Thus, bulk bands were highlighted, and the edge states were indicated by dashed lines.

Reviewer #3 specific remark #5

Why are these interfaces called “adiabatic” and which role does it play?

Authors’ response to Reviewer #3 remark #5

We believe that our choice of the term “adiabatic” is justified as it allows to make mass term profiles and domain wall smooth, i.e., as slowly varying spatially, as needed. The main role of such adiabaticity (smoothness of the topological transition in space) is in the delocalization of the edge modes which brings several benefits, including less sensitivity to the shape of interface and improved radiative lifetime. We hope Reviewer #3 will find our wording well-justified, but we are open to their suggestions about alternative terminology.

Reviewer #3 specific remark #6

Why does this idea do not work for electronic TI and, more generally, for all SPT phases?

Authors’ response to Reviewer #3 remark #6

Before reading comments from Reviewer #1, we too were unaware of any possibility of such adiabatic transitions in electronic systems. The references listed by Reviewer #1, however, show that it is possible to realize smooth transitions in condensed matter systems as well, and Reviewer #1 pointed out some advantages that smooth topological interfaces may bring to electronic topological insulators. We believe that our revised text, and the section that highlights these findings mentioned by Reviewer #1, also shows that the mechanism suggested in our work can potentially be realized in electronic SPT systems.

Reviewer #3 specific concluding remark

In conclusion, while I think the paper is already fine for publication, addressing the points above may serve for even more readability.

Authors’ response to Reviewer #3 concluding remark

We would like to thank Reviewer #3 again for recommending our work for publication and we believe that their suggestions allowed us to further improve its quality and readability.

REVIEWERS' COMMENTS

Reviewer #2 (Remarks to the Author):

The authors have given satisfactory reply to my comments and questions. By amending the weak points, the revised manuscript is now convincing and attractive. I believe it will make a strong case in topological photonics, particularly in showing the genuine robust transport in topological interface channels. This is really an impressive progress. I recommend the paper towards publication as it is.

Reviewer #3 (Remarks to the Author):

The authors have revised their manuscript according to all previous reports. I am satisfied their revisions and response according to my previous report. I would like to recommend it for publication in Nature Communications.

Reviewer #4 (Remarks to the Author):

I am happy with the author's responses and recommend publication of this manuscript in Nature Communications.